# Ultrasonic Vibration-Assisted Stamping of Serpentine Micro-Channel for Titanium Bipolar Plates Used in Proton-Exchange Membrane Fuel Cell

**DOI:** 10.3390/ma16093461

**Published:** 2023-04-28

**Authors:** Yucheng Wang, Qi Zhong, Risheng Hua, Lidong Cheng, Chunju Wang, Haidong He, Feng Chen, Zhenwu Ma

**Affiliations:** 1Robotics and Microsystems Center, School of Mechanical and Electric Engineering, Soochow University, Suzhou 215131, China; 2College of Mechanical Engineering, Suzhou University of Science and Technology, Suzhou 215009, China

**Keywords:** bipolar plates for PEMFC, ultrasonic vibration, micro-channels, acoustic softening, forming limitation

## Abstract

Metallic bipolar plates (BPPs) are key components in the proton-exchange membrane fuel cell (PEMFC), which can replace traditional fossil fuels as a kind of clean energy. However, these kinds of plates, characterized by micro-channels with a high ratio between depth and width, are difficult to fabricate with an ultra-thin metallic sheet. Then, ultrasonic-vibration-assisted stamping is performed considering the acoustic softening effect. Additionally, the influence of various vibration parameters on the forming quality is analyzed. The experimental results show that ultrasonic vibration can obviously increase the channel depth. Among the vibration parameters, the vibration power has the maximum influence on the depth, the vibration interval time is the second, and the vibration duration time is the last. In addition, the rolling direction will affect the channel depth. When the micro-channels are parallel to the rolling direction, the depth of a micro-channel is the largest. This means that the developed ultrasonic-vibration-assisted stamping process is helpful for improving the forming limitation of micro-channels used for the bipolar plates in PEMFC.

## 1. Introduction

Due to the greenhouse effect and rising environmental challenges induced by traditional fossil fuels, the proton-exchange membrane fuel cell (PEMFC) has huge potential application as a clean and renewable energy [1,2,3]. Bipolar plates (BPPs) are key components for a PEMFC stack, and their innovation is required to contribute an additional ~20% improvement in the power density in the following decades [4]. Metallic BPPs possess several advantages in mechanical properties, etc., increase power density, and lower the price when fabricated with an ultra-thin sheet via the stamping process [5]. Further, they are a candidate for utilization in the automotive industry because the volume and weight are critical parameters [6]. However, metallic BPPs in a PEMFC are difficult to fabricate with the decreasing channel width and thickness of the metallic sheet. 

Lots of investigations have focused on innovation in the stamping process to increase the quality of BPPs. By using the rigid-punch stamping and hydroforming process, Koç et al. studied the effect of channel dimensions on the formability and surface quality. Lower variations between plates can be obtained for 3.4%. The surface roughness is increased by the higher hydroforming pressure and stamping force [7]. The fact that the depth of the channel becomes twice as large indicates that the limitation of stamping induced by strain hardening can be improved through heat treatment. For the anisotropy of a rolled sheet, the deeper channel can be stamped in the transverse direction [8]. The shape of the channel also affects the forming depth for the variable curvature and radius of the fillet. An empirical equation is developed to characterize the relationship between reductions in thickness and channel depth, and the fracture can be avoided by controlling the thickness distribution [9]. The accuracy of the stamped channel can be improved by optimizing the forming forces, for example, the error 2.9% of depth uniformity is obtained [10]. With a kind of rigid die, the ultra-thin sheet easily becomes thinner at the corners of the punch and female die due to friction. To increase the depth of microchannels and uniformity of thickness, a flexible forming process with soft mold was carried out. A polymer powder medium is selected as a medium to form BPPs. The depth is increased from 0.4 mm for rigid die forming to 0.6 mm for soft mold, and the thickness uniformity of the BPPs’ wall is increased [11]. Polyurethane rubber, as a soft die, is used to manufacture metallic BPPs, and the effect of its hardness is analyzed. It is found that smaller hardness and bigger thickness lead to a higher filling percentage. Additionally, rubber with micro-features is demonstrated to increase the formability of BPPs [12]. Compared with stainless steel, pure titanium, as a kind of difficulty deformation material, possesses high corrosion resistance and low density, which can help the PEMFC achieve long life and lower weight [13]. However, its maximum elongation is only half that of stainless steel. With the suitable parameters, 68% of channel depth can be punched [14]. Benefitting from the lower density of titanium, the power density of the PEMFC in weight reaches higher than 1.353 kW/Kg [15]. In fact, the channel depth is not deep enough for the PEMFC. To change this situation, a forming process with multi-stages is applied to increase from 438.1 for one-stage stamping to 621 μm. Further, the effect of rolling direction appears for higher depth in the parallel direction [16]. Since the cost of die in forming BPPs is very high, it is necessary to reduce the number of stages for the lower formability of materials. A new process should be developed to improve the formability of ultra-thin sheets. 

Another method is changing the load–time curve, such as static and dynamic force. A square curve application with dynamic force can increase the depth and uniformity of the micro-channels in BPPs more than in static loading conditions, which is helpful in obtaining larger current density [17]. To improve the forming ability, ultrasonic-vibration-assisted forming is carried out for the existing acoustic softening effects [18]. In the deep drawing of a micro-cup, the limited ratio can be increased from 2.58 to 2.94 by using 20 kHz frequency vibration. The utilization of vibration only before max. load is helpful for obtaining better accuracy and depth in micro-cups [19]. Huang’s investigation shows that the vibration amplitude is the most important parameter from the viewpoint of forming limitation [20]. In incremental sheet forming, ultrasonic vibration is useful for decreasing the strain hardening, difference of residual stress and thickness, and increasing the formability when the deformation is small [21,22]. Ultrasonic vibration, as a kind of energy, is used to mold plastic powder as a soft punch, and the bulging ability of the thin sheet could be increased to reach 84% [23]. By using ethylene-vinyl acetate (EVA) soft punch directly, a large part of surface texturing with a thin sheet is bulged to higher micro-cap when ultrasonic vibration is applied, and the complex surface can be precisely fabricated [24,25]. Embossing with rigid punch is also carried out to manufacture micro-features characterizing sine curve, and the springback can be inhibited for the Blaha effect [26]. However, investigations are focused on stainless-steel and non-ferrous metal. The pure titanium is a kind of difficult-to-deform material, and its crystal structure and parameters of material are different from those shown above. Additionally, the forming behaviors have not been studied in ultrasonic-vibration-assisted micro-stamping.

In the study, the ultrasonic-vibration-assisted stamping of micro-channels is performed considering the Blaha effects to investigate its effect on deformation behaviors. The ultra-thin sheet of TA1 pure titanium is selected as a kind of difficult-to-deform material, and the effect of rolling direction is studied. Then, the influence of vibration parameters is analyzed by measuring the depth of micro-channels. The mechanism of ultrasonic vibration improving the forming ability is studied considering the Blaha effect from the viewpoint of acoustic energy density.

## 2. Experimental Setup

The ultrasonic vibrator was the main device in the experiments, which was produced by Shenzhen Rifa Ultrasonic Equipment Co. LTD, as shown in Figure 1a; its maximum power was 2 kW, and the frequency of output vibration was 20 kHz. During the tests, the parameters, such as power, duration time, and so on, were changed using a vibrator controller (Figure 1b). A servo-driven machine, which was developed by our group, was utilized with maximum load 10 kN. All experiments were performed at room temperature with polyethylene (PE) film as lubricant. The displacement of punch was measured using a grating ruler with a precision of 5 μm.

Considering the characteristics of channels in bipolar plates, a kind of serpentine micro-channel was designed, as shown in Figure 2. The width, depth, single angle, and radius of channels at bottom side were 0.9 mm, 0.5 mm, 24°, and 0.25 mm, respectively. In the investigation, the ultra-thin sheet of TA1 was utilized with 0.1 mm in thickness, and its properties are shown in Figure 3. The parameters used in the tests are shown in Table 1. Additionally, the effects of ultrasonic power and duration time were mainly studied by analyzing the depth of stamped micro-channels, measured using a digital microscope (VHX-7000, Keyence, Osaka, Japan). 

## 3. Experimental Results and Analysis

### 3.1. Effect of Rolling Direction of Thin Sheet

Mechanical properties for different directions are obviously different, as shown in Figure 3. To investigate their effects, three kinds of stamping were carried out considering the angle between the rolling direction and channel direction (Figure 4). Additionally, the stamped specimens are shown in Figure 5. From the photograph, it is difficult to find the difference. The outlines of micro-channels were measured using a microscope (Figure 6). To study this in detail, the depth of the stamped channels was calculated, and the experimental results are shown in Figure 7. It can be found that the depth for different directions was different. When the angle was 0°, the micro-channel possessed a maximum depth of 215.54 μm. Additionally, the minimum depth of 184.63 μm was for an angle of 90°. These results indicate that the rolling direction has a clear effect on the stamping forming behaviors, which should be considered in the forming of bipolar plates for a PEMFC with a pure titanium ultra-thin sheet. 

### 3.2. Effect of Ultrasonic Power on the Depth of Micro-Channels

During the stamping of the micro-channel, the punch moves down with a constant speed. From the contacting time between the punch and ultra-thin sheet to the end of the stamping process, it will take about 33 s. From Table 1, three-times ultrasonic vibration will be applied, contacting time, middle of the process, and end of the process, respectively, and the duration time is about 1 s. The effects of ultrasonic power are studied by measuring the outlines and depth of stamped micro-channels, as shown in Figure 8 and Figure 9. The experimental results indicate that the depth of the micro-channel increases with increasing ultrasonic power. Specifically, for power which exceeds 40%, the depth is clearly increased 10 μm when the power increases by 10%. Additionally, for 70% power, the depth is increased to 308 μm, which is about 100 μm bigger than that without ultrasonic vibration. This can be attributed to the acoustic softening effects. With the assistance of ultrasonic vibration, a kind of physical field, the forming properties are improved, which is increased with the ultrasonic power, named the acoustic energy density [21]. 

### 3.3. Effect of Duration Time on the Depth of Micro-Channels

In addition to the power of ultrasonic vibration, the time of ultrasonic vibration applied in the stamping is another parameter, which determines the input energy density. In the investigation, the duration time is selected as 0.5, 0.75, 1.0, 1.25, and 1.5 s, respectively, and the power is 40%, with an interval time of 10 s. The outlines and depths of micro-channels are shown in Figure 10 and Figure 11. With an increase in the duration time, the depth of the micro-channel becomes bigger. Especially from 0.75 s to 1 s, the depth of micro-channels is increased by 35 μm. Compared with that without ultrasonic vibration, the depth of the micro-channel increases about 62 μm for a duration time of 1.5 s. For the “Blaha effects”, the flow stress is decreased immediately when the ultrasonic vibrator is turned on, and it will return to the normal stress when the ultrasonic vibrator is turned off [21]. For smaller flow stress, the depth of the micro-channel is increased. However, since the duration time is very short, its effect on the stamping of the micro-channel is small. Regarding the fracture induced via ultrasonic vibration, the duration time is controlled to be small. 

### 3.4. Effect of Interval Time on the Depth of Micro-Channel

With a constant punch speed, the acoustic energy density of ultrasonic vibration is increased by shortening the interval time. Therefore, its effects are studied by decreasing the interval time from 10 s to 6 s, with 40% power and a duration of 1 s. The outlines and depth of micro-channels are shown in Figure 12 and Figure 13. With the deceasing interval time, the depth of the micro-channel increases. When the interval time is shortened to about 6 s, the increase in depth becomes bigger than that for the bigger one. In fact, for the smaller internal time, three-times ultrasonic force increases to five-times during the stamping of micro-channels. As a result, the depth of the micro-channel is increased by shortening the interval time, which increases the acoustic energy density for the whole stamping process. When the internal time is shortened to the one that is smaller than 6 s, the ultra-thin sheet will be broken for the bigger energy density. 

### 3.5. Effect of Vibration on the Limitation of Micro-Channel’s Depth

For the bipolar plates used in the PEMFC, micro cracks in the micro-channel should be avoided. Then, the effect of vibration on the limitation of the micro-channel’s depth must be studied. In the investigation, the stamped micro-channel is observed after every increase in the punch displacement of 0.02 mm. The depth before fracture is considered as the depth limit of the channel. The depth limit of the micro-channel is shown in Figure 14. It can be seen that the depth limit is increased from 285.5 μm to 311.5 μm using ultrasonic power of 70%, interval time of 10 s, and duration of 1 s, which is induced by the acoustic softening effect. With the assistance of ultrasonic vibration, the uniform deformation of the ultra-thin sheet can be improved by changing the sheet parameters, such as, the *n* value and *r* value [22]. As a result, the depth limit of the micro-channel becomes bigger with ultrasonic vibration.

## 4. Conclusions

In the investigation, an ultrasonic-vibration-assisted stamping process is carried out utilizing ultra-thin sheets of TA1 pure titanium. The effects of ultrasonic vibration on the stamping are analyzed by measuring the depth of micro-channels. The mechanism is studied considering the acoustic softening effects by introducing the energy density. The following conclusions can be obtained.

(1)The rolling direction has a clear effect on the stamping forming process. The max. micro-channel depth of 215.54 μm and the minimum depth of 184.63 μm are obtained for angles of 0° and 90°, respectively.(2)The depth of the micro-channel becomes bigger for the application of higher ultrasonic power, which is about 100 μm bigger for a power of 70% than that without ultrasonic vibration. This can be attributed to the acoustic softening effects.(3)With an increase in the duration time, the depth of the micro-channel becomes bigger. Compared with that without ultrasonic vibration, the depth of the micro-channel increases about 62 μm for a duration of 1.5 s for the “Blaha effects”.(4)With deceasing interval time, the depth of the micro-channel increases. When the interval time is shortened to about 6 s, the increase in depth becomes bigger than that for bigger one, which can be attributed to the increase in the energy density for the whole stamping process.(5)The depth limit is increased from 285.5 μm to 311.5 μm through ultrasonic power of 70%, interval time of 10 s, and duration of 1 s, which can be explained from the viewpoint of the acoustic softening effects. This means that the developed process is helpful for improving the forming limitation of micro-channels used for the bipolar plates in PEMFCs.

## Figures and Tables

**Figure 1 materials-16-03461-f001:**
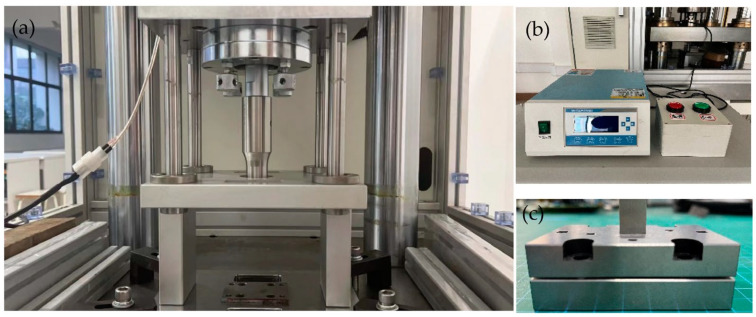
Ultrasonic-vibration-assisted stamping device. (**a**) Experimental device, (**b**) vibrator controller, (**c**) micro mold.

**Figure 2 materials-16-03461-f002:**
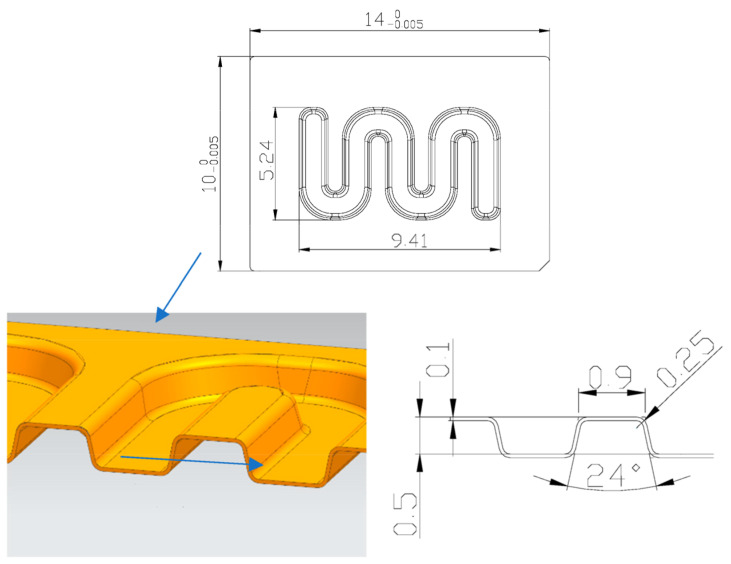
Dimensions of microchannel.

**Figure 3 materials-16-03461-f003:**
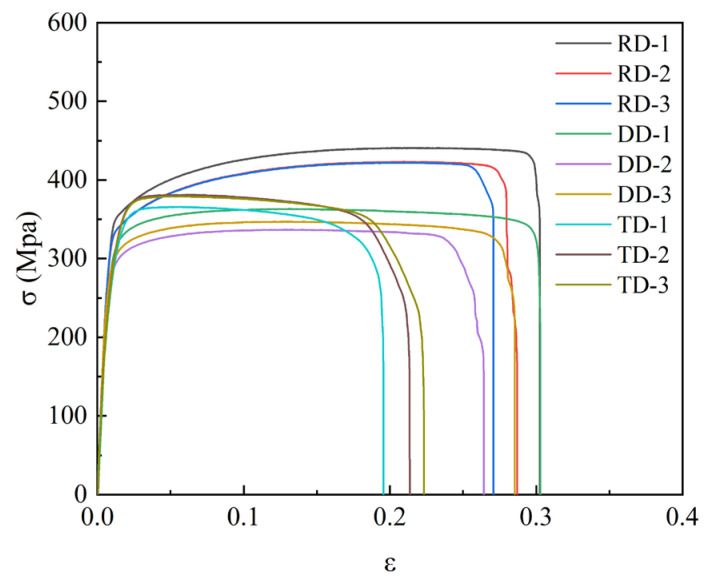
Curves of stress–strain for TA1 properties.

**Figure 4 materials-16-03461-f004:**
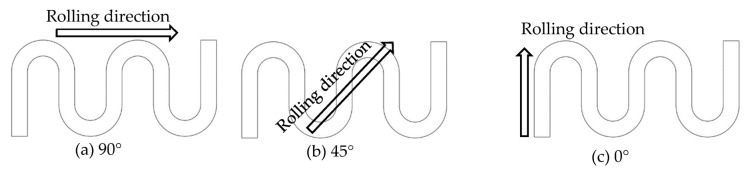
Diagram of rolling direction of sheet in the stamping.

**Figure 5 materials-16-03461-f005:**
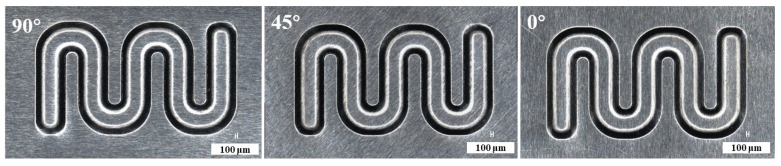
Photograph of stamped channels for different directions.

**Figure 6 materials-16-03461-f006:**
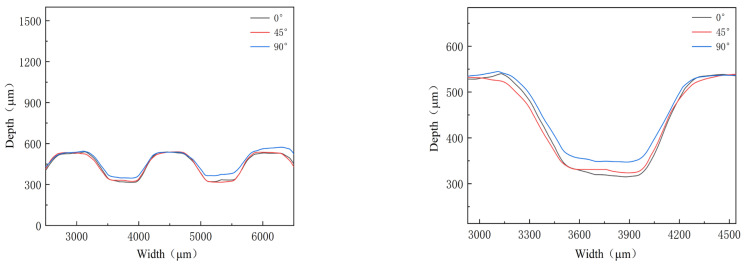
Outlines of micro-channels for different directions.

**Figure 7 materials-16-03461-f007:**
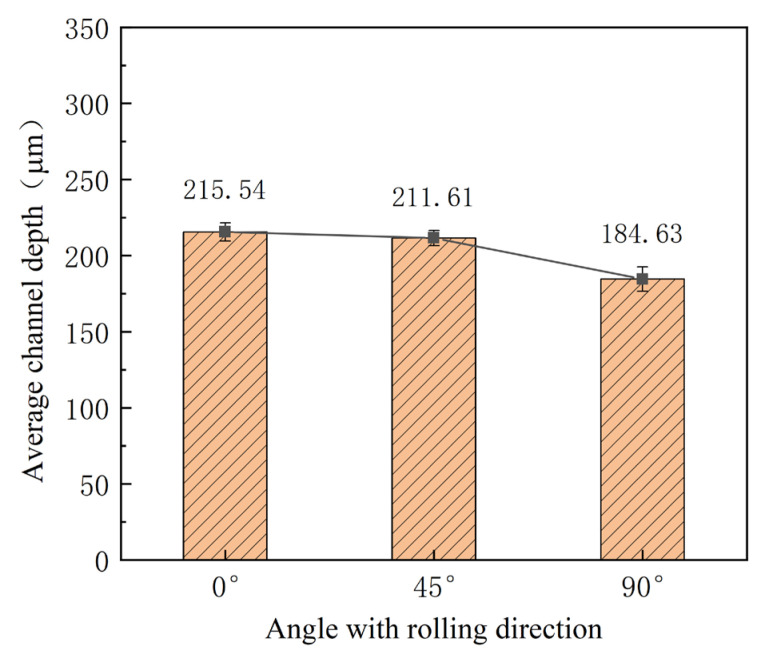
Depth of micro-channels for different directions.

**Figure 8 materials-16-03461-f008:**
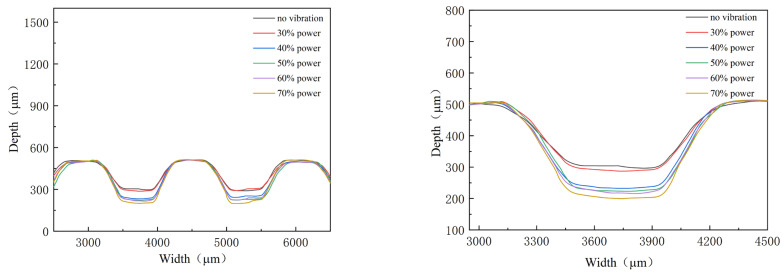
Outlines of micro-channels for different ultrasonic power.

**Figure 9 materials-16-03461-f009:**
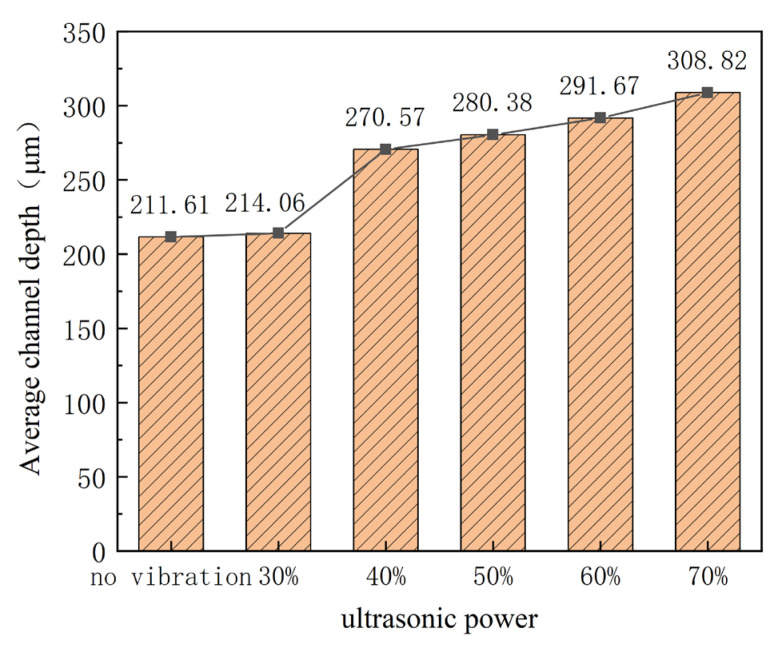
Depth of micro-channel for different ultrasonic power.

**Figure 10 materials-16-03461-f010:**
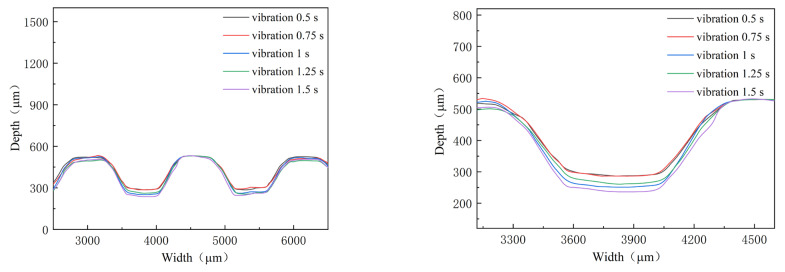
Outlines of micro-channels for different duration time.

**Figure 11 materials-16-03461-f011:**
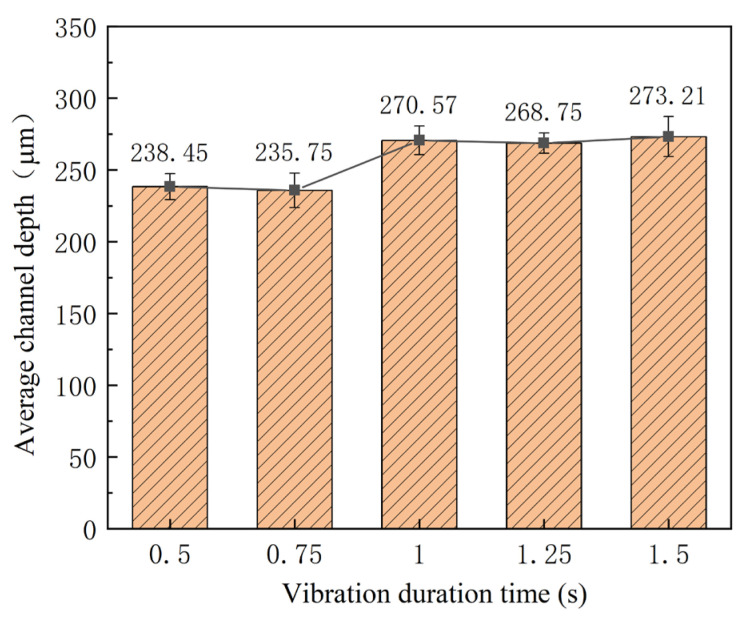
Depth of micro-channel for different duration time.

**Figure 12 materials-16-03461-f012:**
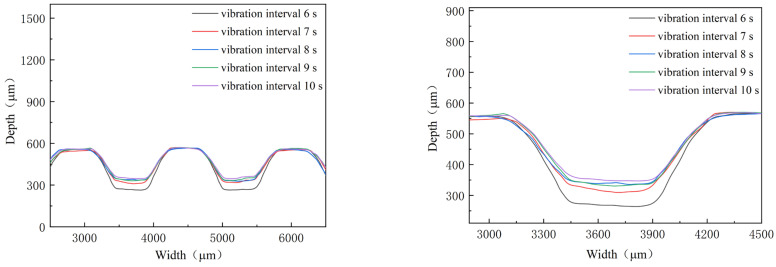
Outlines of micro-channels for different interval time.

**Figure 13 materials-16-03461-f013:**
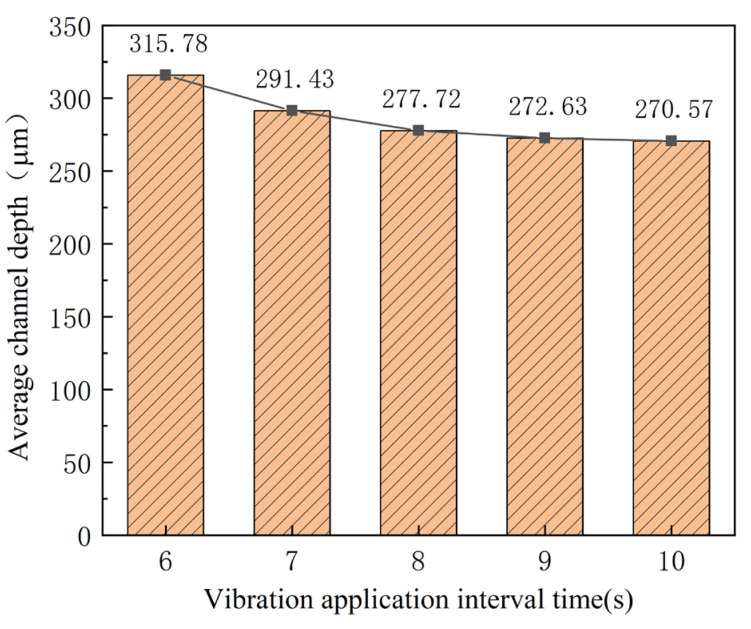
Depth of micro-channels for different interval time.

**Figure 14 materials-16-03461-f014:**
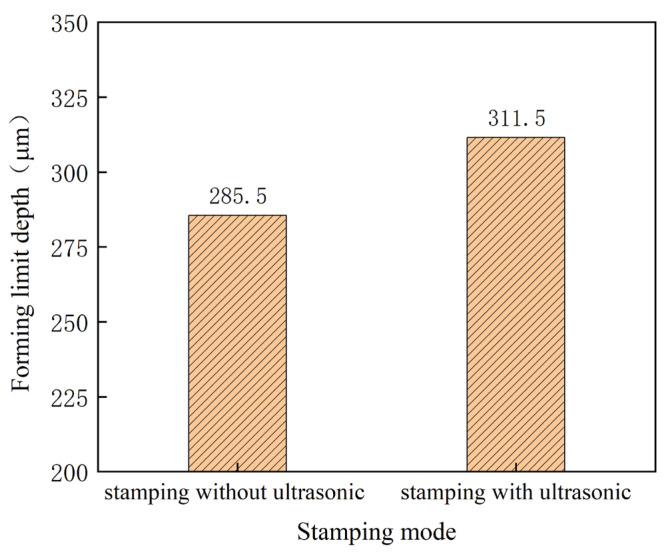
Depth of micro-channels with and without ultrasonic vibration.

**Table 1 materials-16-03461-t001:** Parameters of ultrasonic vibrator used in tests.

Number of Specimen	Power (%)	Duration Time (s)	Interval Time(s)
Without ultrasonic	/	/	/
1	30	1	10
2	40	1	10
3	50	1	10
4	60	1	10
5	70	1	10
6	40	0.5	10
7	40	0.75	10
8	40	1	10
9	40	1.25	10
10	40	1.5	10
11	40	1	10
12	40	1	9
13	40	1	8
14	40	1	7
15	40	1	6

## Data Availability

Not applicable.

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
