# Peer review of "Ultrasonic Vibration-Assisted Stamping of Serpentine Micro-Channel for Titanium Bipolar Plates Used in Proton-Exchange Membrane Fuel Cell"

_materials, 2023, doi:10.3390/ma16093461_

Round 1

Reviewer 1 Report

Overall, the manuscript presents an experimental study investigating the effects of rolling direction, vibration amplitude, application time, and interval time on the stamping process of micro-channels in titanium ultra-thin sheets. The results show that the rolling direction has a clear effect on the depth of micro-channels, and ultrasonic vibration can improve the stamping properties and increase the depth of micro-channels. However, the manuscript should be proof read for the errors for example in the experimental section, subsection 2.3 and 2.4 ( which is again labeled as 2.3) should be correct and so on. 

Author Response

Reviewer 1

Overall, the manuscript presents an experimental study investigating the effects of rolling direction, vibration amplitude, application time, and interval time on the stamping process of micro-channels in titanium ultra-thin sheets. The results show that the rolling direction has a clear effect on the depth of micro-channels, and ultrasonic vibration can improve the stamping properties and increase the depth of micro-channels. However, the manuscript should be proof read for the errors for example in the experimental section, subsection 2.3 and 2.4 ( which is again labeled as 2.3) should be correct and so on. 

Response: Thank you for your comments. The manuscript is carefully revised, and several mistakes are corrected.

Reviewer 2 Report

The manuscript may consider for publication after responding to the following comments and major revising the manuscript properly.

1. Literature review needs to include several recent, relevant publications (high impact) highlighting their key findings. The current version only discussed general aspects while the review of each from several papers is necessary. You may provide a review summary table consisting of a column for the comments or key conclusions.

2. More recent relevant literature or similar work discussion is mandatory in the introduction section, which is missing in the Introduction. Authors are suggested to add one paragraph in the introduction section by discussing the recent progress and citing similar work.

3. The novelty of the work is missing in the introduction. Authors are suggested to include a separate paragraph discussing the novelty and importance of the present work.

4. Authors are suggested to include a literature review on the recent publication by citing following references in the introduction section: DOIs:

1. 10.1016/j.jpowsour.2021.230723
2. 10.1016/j.ceramint.2021.05.167
3. 10.1007/s42247-021-00230-5

5. Provide a more appealing title with no acronyms in a precise and concise manner.

6. Omit trivial information.

7. Explain in brief how the present paper differs from the published ones.

8. State-specific objectives.

9. Provide better-quality figures.

10. State the main findings in the conclusions.

11. Remove less significant and unrelated or less related references and ensure that all the references are cited and arranged sequentially as required by the journal.

English language should be checked throughout the manuscript.

Author Response

Reviewer 2

The manuscript may consider for publication after responding to the following comments and major revising the manuscript properly.

  1. Literature review needs to include several recent, relevant publications (high impact) highlighting their key findings. The current version only discussed general aspects while the review of each from several papers is necessary. You may provide a review summary table consisting of a column for the comments or key conclusions.

Response: Thank you for your comments. Several papers are cited in the manuscript. And, their contributions are summarized in the introduction section. Since investigations on the topic are relatively few, summary table is not prepared in the manuscript.

  1. More recent relevant literature or similar work discussion is mandatory in the introduction section, which is missing in the Introduction. Authors are suggested to add one paragraph in the introduction section by discussing the recent progress and citing similar work.

Response: Thank you for your comments. The recent progress is discussed in the introduction section, and some similar work is cited.

  1. The novelty of the work is missing in the introduction. Authors are suggested to include a separate paragraph discussing the novelty and importance of the present work.

Response: Thank you for your comments. The novelty of the work is provided in the last paragraph of the introduction section.

  1. Authors are suggested to include a literature review on the recent publication by citing following references in the introduction section: DOIs:
  2. 10.1016/j.jpowsour.2021.230723
    2. 10.1016/j.ceramint.2021.05.167
    3. 10.1007/s42247-021-00230-5

Response: Thank you for your comments. The suggested papers are cited in the manuscript.

  1. Provide a more appealing title with no acronyms in a precise and concise manner.

Response: Thank you for your comments. The title is changed to “Ultrasonic vibration assisted stamping of serpentine micro-channel for titanium bipolar plates used in proton-exchange membrane fuel cell”.

  1. Omit trivial information.

Response: Thank you for your comments. Some trivial information is omitted.

  1. Explain in brief how the present paper differs from the published ones.

Response: Thank you for your comments. The shape and dimensions of micro-features are different, and ratio of height-to-width is much bigger than before. Also, pure titanium, as kind of difficulty-deformation material, is selected considering its lower density and higher corrosion resistance.

  1. State-specific objectives.

Response: Thank you for your comments. The aim of the investigation is to develop a new method, such as ultrasonic vibration assisted stamping process, to improve the forming limitation of micro channels with pure titanium ultra-thin sheet used for the bipolar plates in PEMFC.

  1. Provide better-quality figures.

Response: Thank you for your comments. Several figures with high quality are provided in the manuscript.

  1. State the main findings in the conclusions.

Response: Thank you for your comments. The conclusions are revised to state the main findings.

  1. Remove less significant and unrelated or less related references and ensure that all the references are cited and arranged sequentially as required by the journal.

Response: Thank you for your comments. The references are well arranged again, and some new references are added.

Reviewer 3 Report

Please see the file attached.

Please see the above file attached.

Author Response

Reviewer 3

The authors present an interesting experimental work dealing with the investigation on ultrasonic vibration assisted stamping of serpentine micro‐channel used for bipolar plates in proton exchange membrane fuel cell (PEMFC). To investigate the effect on the forming properties of micro‐channels, the ultrasonic assisted stamping is performed considering the Blaha effects. The ultra‐thin sheet of TA1 pure titanium is selected as a kind of light weight material, and the effect of rolling direction is studied. Then, the influence of various vibration parameters on the depth is analyzed by measuring the depth of micro‐channel. The mechanism is studied considering the “Blaha” effect from the viewpoint of the acoustic energy density.

My general comments are listed as follows:

 1. The authors should directly mention the novelty and contribution of their work. Ideally the authors should mention their added value either in abstract or in the last sentence of the introduction section.

Response: Thank you for your very good advice. Contribution of some works is added in the introduction section.

 2. The text should be checked for grammar and spelling, typos, etc. See for example in page 2 of 12, section 2 “Experimental setup” 6th line, the word “maximum” has been written as “maxmum”.

Response: Thank you for your comments. The manuscript is carefully checked, and several mistakes are corrected.

 3. All abbreviated words should be directly given once they are appeared, i.e., “PE film as lubricant” page 2 of 12, section 2 “Experimental setup” 6th line.

Response: Thank you for your comments. The full names are given for some abbreviated words.

 4. Some values for axes in experimental results in figures should be enlarged to facilitate reading. See for example the stress‐strain diagram for TA1 material properties in Figure 3. The same also goes for figure 5 (Photograph of stamped channels for different directions). The measuring scale appears to be 100.00μm (see down‐right corner of sub‐figures, 90deg., 45deg., and 0deg.) yet it is hardly readable.

Response: Thank you for your comments. The words in the several figures are enlarged.

 5. The authors should briefly explain why the authors decided to select the values presented in Table 1 (page 4 of 12) for their experiments.

Response: Thank you for your comments. Before we select values in the Table 1, some exploratory experiments are conducted. Based on the results, we decided to select the values as shown in Table 1.

 6. It would be quite interesting for the authors to setup a systematic design of experiments (i.e. Taguchi orthogonal arrays or response surface methodology) and try to find out the effect of process parameters (as well as their interactions) on discrete and important responses.

Response: Thank you for your very good advice. Experiments in the manuscript are designed to discover the mechanism of ultrasonic vibration as a kind of energy field on the plastic deformation, by considering the acoustic softening effect. In the following investigation, we will adopt your advice to setup a systematic design of experiments.

Round 2

Reviewer 1 Report

The authors have sufficiently improved the manuscript and should be accepted in present format. 

Reviewer 2 Report

Accept

Accept

Reviewer 3 Report

The authors have properly revised their interesting work presented in their manuscript. Therefore i would recommend the acceptance of the revised manuscript.